# Intensity vs. Volume in Professional Soccer: Comparing Congested and Non-Congested Periods in Competitive and Training Contexts Using Worst-Case Scenarios

**DOI:** 10.3390/sports13030070

**Published:** 2025-02-27

**Authors:** Tom Douchet, Antoine Michel, Julien Verdier, Nicolas Babault, Marius Gosset, Benoit Delaval

**Affiliations:** 1Racing Club de Lens (RCL), 33 Rue Arthur Lamendin, F-62210 Avion, France; tom.douchet@gmail.com (T.D.); mgosset@rclens.fr (M.G.);; 2INSERM UMR1093-Cognition Action et Plasiticité Sensorimotrice, Université Bourgogne Europe, UFR des Sciences du Sport, F-21000 Dijon, France; 3Centre d’Expertise de la Performance, Université Bourgogne Europe, UFR des Sciences du Sport, F-21000 Dijon, France

**Keywords:** load management, physical demands, performance monitoring, integrated training, player workload, recovery strategies

## Abstract

**Background:** Understanding the balance between intensity and volume during training and competition is crucial for optimizing players’ performance and recovery in professional soccer. While worst-case scenarios (WCSs) are commonly used to assess peak match demands, little is known about how the time spent within WCS thresholds varies across congested and non-congested periods, especially when considering differences in playing time. This study examines the time spent at different percentages of WCSs during congested and non-congested periods for players with lower and higher playing times throughout training sessions and matches. **Methods:** Data were collected from a professional soccer team across a congested and non-congested match period. Twenty players were divided into two groups based on playing time: the top 10 playing times (PT 1–10) and the bottom 10 playing times (PT 11–20). WCS thresholds for total distance (TD) and the distance covered above 20 km·h^−1^ (D20) were quantified in 10% increments, starting from 50% and increasing up to >100%. The time spent at each threshold was compared between periods and groups for the integrated soccer exercises performed during all training sessions. Repeated measures of ANOVA were used to analyze differences between playing time groups and periods. **Results:** During training, players spent significantly more time within the 50–90% WCS TD and WCS D20 thresholds during non-congested periods compared to congested periods (*p* < 0.05). However, no significant differences were observed in the time spent for >90% of the WCSs between periods (*p* > 0.05). Both PT 1–10 and PT 11–20 groups exhibited similar patterns of WCS achievement, with small effect sizes observed for a few indicators. **Conclusion:** Coaches should design training sessions that replicate or exceed match demands, particularly during non-congested periods. Future strategies should integrate larger-sided games with longer durations and dissociated contents to better individualize and optimize training loads, especially for non-starters.

## 1. Introduction

One of the primary objectives of training in sports is to replicate or even exceed competitive demands (physical, technical, tactical) to enhance athletes’ performance [1,2,3]. Achieving this requires a precise understanding of competitive demands. Specifically, knowledge of maximum intensities during matches theoretically allows for the calibration of training intensity. While quantifying maximum intensity in terms of internal load is relatively straightforward (e.g., using maximum heart rate) [4], assessing the maximal external load is more complex. External load refers to the physical demands placed on the body, such as the total distance covered, speed, acceleration, or deceleration during training or competition, whereas internal load represents the physiological responses to those demands, typically measured through heart rate, the rate of perceived exertion (RPE), or other physiological markers [5]. Some researchers have proposed using maximal running velocity to quantify the distances covered at various intensities (percentages of maximal velocity) [6]. However, competitive maximal velocity might be underestimated compared to an athlete’s true sprinting performance due to the infrequent occurrence of such actions during matches [2]. Consequently, this discrepancy underscores the need for activity-specific indicators.

To address this, numerous studies have sought to capture the most intense periods of matches. Known as the worst-case scenario (WCS) [7,8,9], this indicator represents the maximum distance covered by an athlete over various time windows. The quantification of competitive WCSs has been shown to be influenced by the match contextual variables. Significant differences in WCSs have been observed across player positions, match halves, match locations, and match outcomes [10]. A recent study proposed quantifying different percentages of the WCS (e.g., 80% and 90%) to better assess the time spent within these thresholds during competition [11]. This simple approach demonstrates that two individuals with identical competitive WCS values may have significantly different durations beyond 80% and 90% of WCSs.

Given the importance of the WCS in assessing the most intense periods of competition, it is essential to understand how these high-intensity demands fluctuate under different match constraints. One key factor influencing external match loads is fixture congestion, which may alter players’ abilities to sustain peak intensities over repeated matches. The impact of congested fixture periods on external match loads remains complex and appears to vary depending on the metrics analyzed. A decrease was observed in certain external load variables, including the total distance covered and accelerometry-based indices, suggesting that players adapt their physical output to cope with increased match frequency [12]. In contrast, previous research reported no significant differences in the total distance covered or high-intensity efforts between congested and non-congested periods, indicating relative stability in physical performance despite fixture accumulation [13]. Similarly, it was found that while the total distance covered and low-intensity running fluctuated over a prolonged congested period, high-intensity efforts remained relatively stable [14]. These findings suggest that while players may adjust their overall running volume to manage workload, their ability to produce high-intensity efforts in matches appears to be maintained, likely due to recovery strategies and squad rotation implemented by the coaching staff.

While high-intensity efforts appear to be maintained during congested fixture periods, the distribution of playing time within a squad introduces another layer of complexity. Unpublished data from the authors indicate that the ten most frequently used players typically account for approximately 70% of the total playing time, while 16 players collectively share around 95% of it (Appendix A). In a squad of 25 to 30 players, this means that some individuals receive little to no match exposure. This discrepancy raises important questions about whether players with the highest playing time experience similar external loads during training sessions compared to those with limited match involvement.

In this context, we sought to investigate whether using the WCS quantification (the time spent within specific thresholds) during training could be used to evaluate the replication of match intensities and determine if playing time influences this quantification. Our objective was to analyze all matches and integrated training exercises across congested and non-congested periods to examine the impact of competitive playing time on the time spent within different WCS thresholds. We hypothesized that players with reduced playing time would exhibit longer durations within WCS thresholds during training, while starters would have higher values during competition.

## 2. Materials and Methods

Twenty-three professional soccer players (age: 26.6 ± 4.2 years, height: 180.0 ± 6.5 cm, body mass: 78.4 ± 7.1 kg) from the first team of a professional French club (Ligue 1) were recruited for this study. Goalkeepers were excluded due to the unique nature of their activity. Therefore, the sample included 5 central defenders, 5 full-backs, 6 central midfielders, 3 wide midfielders, and 4 forwards. Only players who completed every training session during the protocol and who played for at least five competitive minutes were included in the study. Data were obtained from the club throughout the 2024–2025 season, as players were routinely monitored during training and matches. Therefore, ethics approval was waived. Nevertheless, to guarantee team and player confidentiality, all data were anonymized before analysis. To ensure data quality, all datasets were screened for missing values or inconsistencies. Extreme outliers (>3 standard deviations from the mean) were removed unless a contextual justification was identified. Only complete datasets from fully tracked sessions were included in the analysis. The research was conducted in accordance with the Declaration of Helsinki.

### 2.1. General Design

The study was conducted across two distinct periods: a congested period followed by an international break (8 days) and a subsequent non-congested period (Figure 1). The congested period spanned 14 days and included 4 competitive matches and 10 training sessions. Between matches, only 2 or 3 training sessions were held. Training sessions during this period adhered to a specific structure: a recovery-oriented compensation session three days before the match (MD-3), a neuromuscular-oriented session two days before the match (MD-2) using mainly small-sided games, and a typical soccer tapering session on the day preceding the match (MD-1) using reactivity and small-sided games [15]. In the case where only 2 training sessions were held, the recovery-oriented session replaced the neuromuscular-oriented session on MD-2. Over the course of this congested period, 20 players were used during competitive games, with 15 different players starting matches and 14 changes in the starting 11.

The non-congested period similarly included 4 competitive matches but spanned 27 days, allowing 19 training sessions (Figure 1). Between matches, there were a maximum of 5 and a minimum of 4 training sessions. Training during this period followed a typical tactical periodization model commonly employed in professional soccer [1]. The structure of this model follows the principles of tactical periodization, where each session is designed to target specific aspects of player performance. The day following the match (MD+1), the recovery session facilitated physical recovery to prepare players for the next session. On MD-4, neuromuscular development is emphasized through small-sided games, focusing on physical conditioning. On MD-3, the aerobic development session uses large-sided games to mimic match intensity. The speed-focused session on MD-2 addresses explosive power and acceleration through specific speed work. Finally, the tapering session on MD-1 aims to optimize reactivity and match-specific readiness through tactical exercises and small-sided games. In the case where only 4 training sessions were held, the recovery-oriented session was conducted on MD-4 for the starters, while the non-starters performed the neuromuscular-oriented session. During the non-congested period, a total of 20 players were used, with 16 different players starting matches and 9 changes in the starting 11.

For each period, we divided the players into two groups: the first group included those with the top 10 playing times (PT 1–10), while the second group consisted of players ranked 11th to 20th in terms of playing time (PT 11–20).

### 2.2. Testing Procedure

During all training and match sessions, players were equipped with a Catapult Vector S7 10 Hz GPS unit integrated with a 100 Hz triaxial accelerometer microsensor (Vector S7, Catapult Innovations, Melbourne, Australia). This system has been validated for reliability in team sports [16]. The typical measurement error for GPS units in team sports is estimated to be approximately 1–2% for the total distance and around 5% for high-speed running distance [17,18]. These values have been confirmed by a comparison between the Catapult Vector S7 system and an optical tracking system (Tracab 25 Hz), which revealed an excellent agreement for total distance (ICC = 0.974) and a good agreement for high-speed running distance (ICC = 0.766) [17]. To minimize inter-unit variability, each player used the same individual GPS device throughout the study, which is in line with best practices to ensure data consistency and reduce measurement errors [18]. The device was positioned between the scapulae using a specialized vest. In accordance with the manufacturer’s recommendations, all devices were activated 15 min prior to data collection to ensure optimal satellite connectivity and data accuracy. GPS units were distributed to players 10 min before the start of each training session. To minimize inter-unit variability, each player consistently wore the same GPS device throughout the study. Devices were deactivated immediately after the conclusion of each training session or match.

Data were calibrated in real time during sessions using a dedicated tablet application, and this was subsequently adjusted with the manufacturer’s software (OpenField v1.17.0), downloaded, and analyzed via the manufacturer’s API (Application Programming Interface (version 0.0.0.60)) in the R statistical environment (version 2024.12.1). After each match, locomotor variables were calculated using raw data. The locomotor variables of interest were the total distance (TD) and high-speed running distance (D20), defined as the distance covered above 20 km·h^−1^.

Two custom-made algorithms were developed in Python (version 3.9.19) for this study. The first algorithm identified the WCS by calculating rolling averages for a 1 min time window, as described before [11,19]. The maximum value was then used to determine one WCS for TD and one for D20, representing the maximum distance covered in TD and D20 over one minute [10]. The second algorithm quantified the cumulative time spent at various intensities, expressed as percentages of each WCS separately (ranging from 50% to 100% of the WCS, with increments of 10%). During training sessions, only integrated soccer training activities were included in the analysis, with dissociated exercises excluded (e.g., all activities without a ball), as they do not reflect match-specific demands.

The same algorithms were applied across both experimental periods (congested and non-congested periods), ensuring consistent methodology. As such, the WCS was independently calculated and saved for each period.

### 2.3. Statistical Analysis

Descriptive statistics for all variables were calculated for the entire group using means and standard deviations. To assess the interaction between the periods (congested vs. non-congested periods) and playing time (PT 1–10 vs. PT 11–20), a two-way repeated measure analysis of variance (ANOVA) was performed. Prior to running the test, the assumption of sphericity was evaluated using Mauchly’s test. For the ANOVA, partial eta-squared (*ηp*^2^) was calculated as a measure of effect size. Effect sizes were interpreted as small (<0.06), moderate (0.06−0.15), or large (>0.15) [20]. Statistical significance was set at *p* < 0.05.

In the event of significant main effects for the period or playing time or a significant interaction between these factors, Bonferroni-adjusted post hoc tests were used to identify localized differences while controlling for multiple comparisons. Mean values are presented alongside the effect sizes (Cohen’s d) and their associated 95% confidence intervals. Effect sizes were interpreted as trivial (<0.2), small (0.2−0.5), moderate (0.5−0.8), or large (>0.8) [16].

## 3. Results

Out of the 23 players recruited for this study, 20 (transfer *n* = 1, injuries *n* = 2) were used for data analyses.

Results from the ANOVA demonstrated no significant interactions between the periods (congested vs. non-congested periods) and the playing time (PT 1–10 vs. PT 11–20) (Table 1).

Significant period effects (congested vs. non-congested periods) were observed (Table 1 and Table 2). During training sessions, the time spent was between 50 and 60% (*p* < 0.001), 60 and 70% (*p* < 0.001), 70 and 80% (*p* < 0.001), and 80 and 90% (*p* = 0.004) for WCS TD, which demonstrated greater values during the non-congested compared to the congested period. Conversely, for WCS D20, the time spent was between 50 and 60% (*p* < 0.001), 60 and 70% (*p* = 0.002), 70 and 80% (*p* = 0.030), and 80 and 90% (*p* = 0.040), and was greater during the non-congested period rather than the congested period. During competitive games, no significant differences were shown for WCS TD. However, for WCS D20, the non-congested period demonstrated a greater time spent at 50–60% (*p* = 0.002), 60–70% (*p* = 0.048), and 70–80% (*p* = 0.029) than the congested period.

Figure 2 and Figure 3 show the results for WCS TD and WCS D20 for the playing time during training sessions. The playing time parameter (PT 1–10 vs. PT 11–20) did not show any significant difference for WCS TD (Table 1). However, for WCS D20, playing time differentiated for the time spent, which was between 80 and 90% (*p* = 0.023, d = −0.471, 95% CI = −0.916 to −0.026) and 90 and 100% (*p* = 0.042, d = −0.595, 95% CI = −1.204 to 0.015) during training sessions with greater values for PT 11–20 (Table 1). For WCS D20, during competitive games, greater values were obtained for PT 11–20, which were observed between 50 and 60% (7.65 ± 3.07 vs. 2.08 ± 11.36, *p* = 0.002, d = 1.513, 95% CI = 0.462 to 2.564), 60 and 70% (3.56 ± 1.59 vs. 2.13 ± 1.49, *p* = 0.048, d = 0.714, 95% CI = −0.033 to 1.462), and 70 and 80% (2.20 ± 1.35 vs. 1.20 ± 1.19, *p* = 0.029, d = 0.688, 95% CI = 0.031 to 1.345) while no difference was observed between 80 and 90% (1.05 ± 1.01 vs. 0.78 ± 0.69, *p* = 0.689, d = 0.103, 95% CI = −0.117 to 0.345) and 90 and 100% (0.80 ± 0.68 vs. 0.48 ± 0.35, *p* = 0.173, d = 0.183, 95% CI = −0.073 to 0.645).

## 4. Discussion

The aim of our study was to investigate the time spent within specific thresholds of the competitive WCS during congested and non-congested periods, comparing starters and non-starters using competitive games and integrated training exercises. While we hypothesized that players with a reduced playing time would spend more time within various thresholds of the competitive WCS during training sessions, our results do not support this hypothesis, as no interaction between playing time and period was observed. This unexpected result suggests that the current training structure may not sufficiently differentiate workload based on playing time, highlighting the need for further investigation into alternative programming strategies.

The analysis of the congested and non-congested periods revealed that, during training sessions, players spent significantly more time within the 50–60%, 60–70%, 70–80%, and 80–90% thresholds for WCS TD and WCS D20 (Table 1 and Table 2). However, the time spent in the 90–100% and >100% thresholds for WCS TD and WCS D20 did not differ significantly between the congested and non-congested periods. The differences observed in WCS TD and D20 can be explained by the higher number of training sessions during the non-congested period (n = 19) compared to the congested period (n = 10). However, the time spent in the 50–60% to 80–90% thresholds was more than double (WCS TD: 212.1 ± 70.5 min vs. 76.4 ± 35.1 min—WCS D20: 11.4 ± 10 min vs. 3 ± 4.2 min), indicating that not only the number of sessions but also the specific training content influenced the workload experienced by the players.

During the congested period, training content is primarily designed to facilitate players’ recovery from previous matches and prepare for upcoming fixtures [21,22]. In contrast, the non-congested period allows for a more structured tactical-periodization programming approach. This typically includes a recovery-oriented phase (up to MD+2/MD-5) [23], followed by a development phase (MD-4 and MD-3), which is absent during congested periods. This concludes with a tapering phase before the next match (MD-2 and MD-1) [24]. The development phase, particularly on MD-4 and MD-3, seeks to replicate or even exceed competitive demands to enhance players’ physical capacities. These sessions often include small-to-large-sided games with varying area-per-player ratios that address all components of the game [25]. Large-sided games, typically scheduled on MD-3, closely replicate match conditions and play a crucial role in high-intensity training during the week [26]. However, our results show that the time spent at >90% for both WCSs (90–100% and >100%) does not differ significantly between congested and non-congested periods. This suggests that, despite their intended purpose, these sessions may not fully expose players to the highest intensities of match play. A possible explanation for this is that even when external loads appear to be matched, internal load responses—such as neuromuscular fatigue or metabolic strain—could vary, affecting overall adaptation.

Further results showed that playing time did not significantly affect the workload achieved during training sessions, apart from a few indicators with small effect sizes. One possible explanation lies in the nature of top-up sessions designed for players with reduced playing times [27]. While players with lower playing times often participate in post-match top-up sessions, these sessions exclusively involve dissociated drills for practicality. During the week, integrated drills, primarily in the form of small-sided games, are more commonly used for these players [28]. However, these sessions are typically limited to 10–12 players, making it difficult to fully replicate competitive demands, which would ideally require a larger group of 20 players.

Small-sided games can achieve competitive intensities in terms of internal load (e.g., maximal or mean heart rate) [29]. However, replicating external load metrics, such as the percentages of maximal velocity or WCS intensity, could be more challenging. The smaller playing areas used in small-sided games, designed to maintain optimal technical and tactical cohesion, limit the total distance and the distances covered at high intensity per minute [3,30]. Although research has occasionally shown that small-sided games can exceed competitive demands, sustaining such intensities over sufficient volumes (time and distances) to recreate an appropriate intensity load remains challenging [31]. Moreover, our findings suggest that simply mirroring match demands in training may not be sufficient for optimal adaptation. The principle of progressive overload must also be considered, ensuring that training loads progressively challenge players beyond match intensities over time to promote continued adaptations.

These findings raise questions about the necessity of incorporating dissociated drills for non-starters to increase physiological demands. Our results demonstrate that current top-up strategies are insufficient to differentiate between players with high (PT 1–10) and low (PT 11–20) playing times. To address this, two main strategies should be considered. The first one is to incorporate dissociated drills [25]. Dissociated drills offer a practical solution to elevate both internal and external loads. They can be tailored to players’ individual capacities, such as VO_2_max, maximal aerobic speed, maximal heart rate, maximal velocity, or competition-derived metrics, thereby adhering to the training principle of individualization.

The second strategy is to enhance specificity with integrated drills. Specificity, a fundamental training principle, requires soccer-specific activities. However, our results suggest that current integrated approaches fail to differentiate workloads based on playing time. Future research should investigate whether modifying the intensity and duration of these drills can better align training stimuli with adaptation principles, particularly for players needing additional exposure to high-intensity efforts. To overcome this, top-up sessions for players with lower playing times should be designed to function as developmental sessions that replicate match demands (such as MD-3) [21]. This would necessitate larger group sizes to facilitate large-sided games. Professional teams could include reserve team players to increase the number of participants, thereby enabling sessions that better replicate competition demands and elevate the physiological loads of players with lower playing times.

This study offers valuable insights into the workload achieved during training across congested and non-congested periods for players with both lower and higher playing times. However, several limitations need to be addressed. First, the findings of this study are based on data from a single professional team at a specific time of the year, which may limit their generalizability. Future research should aim to replicate these results in different contexts, such as with youth players focused on development rather than competitive outcomes or in different countries with varying methodologies. Furthermore, this study should be replicated during mid-season or at the end of the season. This would provide a broader perspective and account for varying training philosophies across different teams and levels of play.

Second, while the method of quantifying time spent in different WCS thresholds provides a novel and meaningful approach to understanding external load, it represents only one aspect of training demands. It does not capture players’ physiological responses to these loads, which could be better assessed through internal load markers. Integrating measures of neuromuscular fatigue and recovery status in future studies could help clarify whether current training designs adequately stimulate adaptation, particularly for players with reduced playing times.

Finally, although the effect sizes observed in our study are generally small, it is important to consider their practical implications within an elite soccer environment. Even minor differences in training load exposure can contribute to cumulative adaptations over a season, influence recovery strategies, and impact long-term player development. Understanding how different training stimuli interact over time, rather than focusing solely on immediate match-like intensities, could offer a more holistic perspective on performance optimization.

## 5. Conclusions

This study provides valuable insights into the intensity profiles of training sessions across congested and non-congested periods, highlighting key distinctions between volume (time, distance) and intensity (near to maximal competitive demands) in the training loads of players with varying playing times. The findings emphasize the critical role of training content and structure in replicating competitive demands, particularly during non-congested periods, where development-oriented sessions allow for greater exposure to high-intensity thresholds. Conversely, the lack of differentiation in intensity profiles between players with higher or lower playing times suggests the need to optimize top-up strategies, particularly for players with lower playing times, to ensure their readiness for competition. Additionally, the lack of differentiation in training loads could lead to over- or undertraining, increasing the risk of injuries. This study, therefore, has important implications for workload management in relation to injury prevention and the overall health of soccer players.

While this research provides meaningful insights, it also underscores the necessity for further exploration across different team settings and levels of play to validate the applicability and generalizability of the findings. Additionally, integrating internal load metrics with the external load framework could enhance our understanding of player responses to varying training intensities. By addressing these aspects, future research can better inform the design of training programs that are both individualized and aligned with the physiological demands of professional soccer. Rather than considering volume and intensity as separate entities, future research should focus on quantifying the volume of intensity.

## Figures and Tables

**Figure 1 sports-13-00070-f001:**
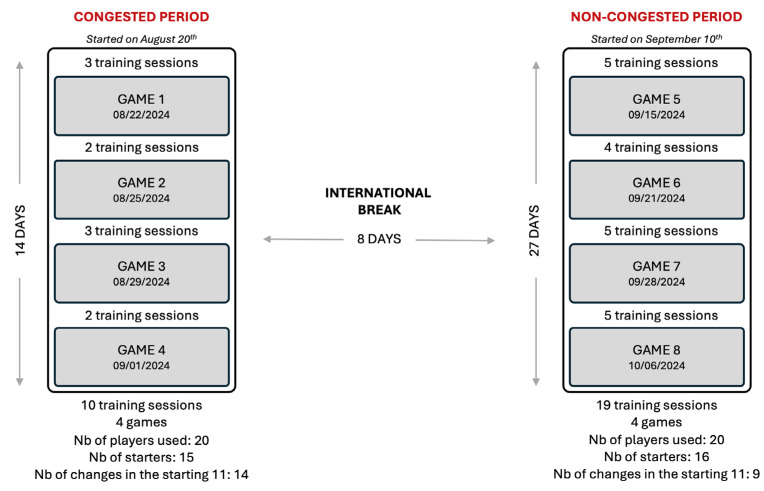
Study flowchart illustrating the characteristics of congested and non-congested periods. The number of training sessions and games are shown. The total number of players, the total number of different starters, and the number of changes in the starting eleven throughout all the competitive games of each period are shown.

**Figure 2 sports-13-00070-f002:**
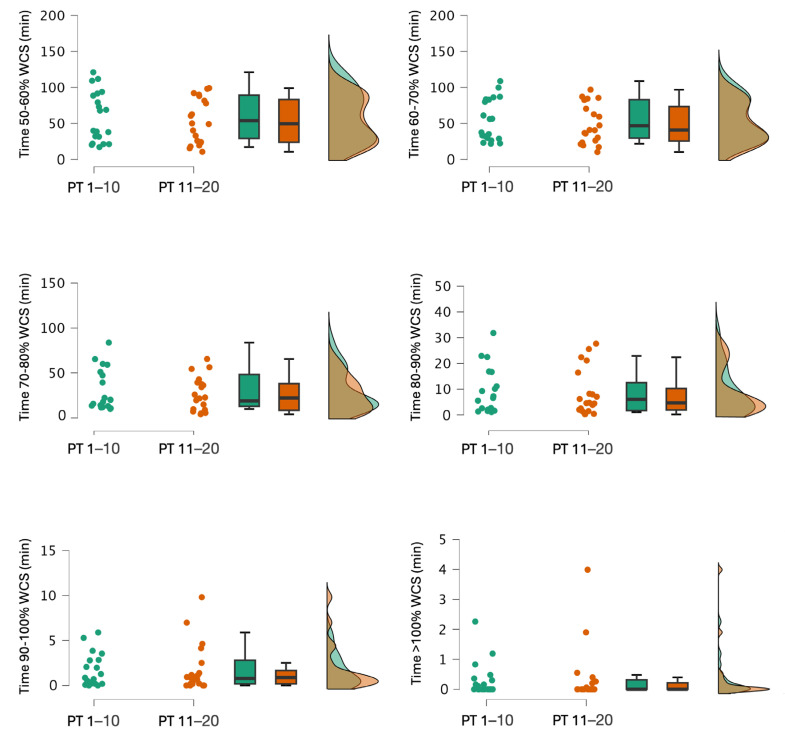
Comparison of the total time spent in all worst-case scenario (WCSs) for total distance (TD) thresholds during training sessions between two groups based on playing time: PT 1–10 (players with the ten highest playing time) and PT 11–20 (players with the ten lowest playing time).

**Figure 3 sports-13-00070-f003:**
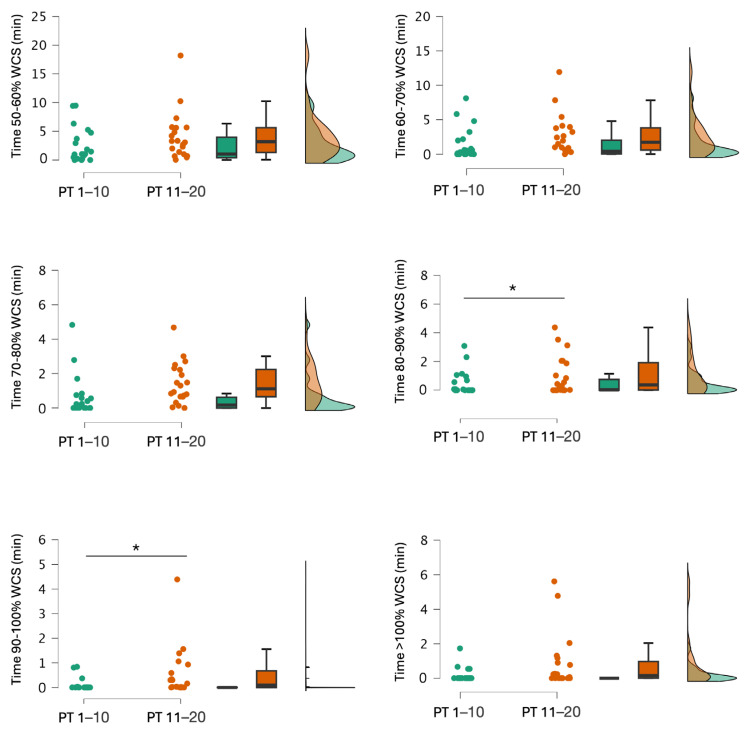
Comparison of the total time spent in all worst-case scenarios (WCSs) for distance covered threshold of >20 km·h^−1^ (D20) during training sessions between the two playing time groups: PT 1–10 (players with the ten highest playing time) and PT 11–20 (players with the ten lowest playing time). Significant differences (*p* < 0.05) are indicated by *.

**Table 1 sports-13-00070-t001:** Results from the mixed model ANOVA for the total time spent on percentages of the WCS TD and WCS D20.

		TD			D20		
		F	*p*	*η* _p_ ^2^	F	*p*	*η* _p_ ^2^
Threshold	Effect	Training Sessions					
50–60%	Period	143.365	<0.001 ***	0.827	24.579	<0.001 ***	0.390
	Playing Time	1.214	0.299	0.010	2.571	0.143	0.055
	Period x Playing Time	1.177	0.306	0.004	0.226	0.646	0.005
60–70%	Period	94.936	<0.001 ***	0.792	19.485	0.002 **	0.299
	Playing Time	2.388	0.157	0.012	2.301	0.164	0.064
	Period x Playing Time	0.959	0.353	0.007	0.001	0.999	0.001
70–80%	Period	28.120	<0.001 ***	0.459	6.578	0.030 *	0.176
	Playing Time	0.760	0.406	0.014	4.615	0.060	0.145
	Period x Playing Time	2.521	0.147	0.048	1.145	0.312	0.018
80–90%	Period	14.578	0.004 **	0.267	5.789	0.040 *	0.164
	Playing Time	0.033	0.860	0.001	7.470	0.023 *	0.074
	Period x Playing Time	0.947	0.356	0.035	0.001	0.990	0.001
90–100%	Period	3.204	0.107	0.041	0.023	0.883	0.01
	Playing Time	0.066	0.803	0.002	5.579	0.042 *	0.118
	Period x Playing Time	1.465	0.257	0.075	0.641	0.444	0.031
>100%	Period	0.176	0.685	0.005	0.490	0.502	0.017
	Playing Time	0.106	0.752	0.003	1.677	0.228	0.053
	Period x Playing Time	4.343	0.067	0.153	0.716	0.419	0.025
Threshold	Effect	Competitive Games					
50–60%	Period	0.285	0.606	0.004	2.565	0.144	0.037
	Playing Time	3.095	0.112	0.198	18.778	0.002 *	0.453
	Period x Playing Time	0.003	0.960	0.001	20.823	0.078	0.093
60–70%	Period	0.060	0.812	0.001	0.780	0.400	0.029
	Playing Time	2.656	0.138	0.166	5.238	0.048 *	0.179
	Period x Playing Time	0.379	0.553	0.005	0.780	0.400	0.012
70–80%	Period	0.001	0.992	0.001	0.597	0.459	0.025
	Playing Time	1.859	0.206	0.109	6.718	0.029 *	0.156
	Period x Playing Time	1.393	0.268	0.030	0.013	0.911	0.001
80–90%	Period	0.011	0.919	0.001	0.837	0.384	0.039
	Playing Time	2.422	0.154	0.114	0.171	0.689	0.006
	Period x Playing Time	0.888	0.371	0.031	0.618	0.452	0.016
90–100%	Period	0.419	0.534	0.015	1.556	0.244	0.064
	Playing Time	5.053	0.051	0.168	2.194	0.173	0.037
	Period x Playing Time	2.023	0.189	0.035	0.672	0.433	0.026

* = *p* < 0.05; ** = *p* < 0.01; *** = *p* < 0.001. TD: total distance; D20: distance > 20 km·h^−1^.

**Table 2 sports-13-00070-t002:** Total time spent in all WCS thresholds for TD and D20 during congested and non-congested periods.

	TD			
	C	NC	d	95% CI
**Threshold**	**Training Sessions**			
50–60%	26.9 ± 10.3	85.1 ± 18.3	−2.978	−4.007 to −1.934
60–70%	28.3 ± 9.0	75.0 ± 19.9	−2.226	−3.045 to −1.390
70–80%	16.1 ± 10.1	39.8 ± 22.3	−1.089	−1.637 to −0.523
80–90%	5.1 ± 5.7	12.2 ± 10.0	−0.688	−1.170 to −0.192
90–100%	1.4 ± 2.3	2.1 ± 2.2	−0.245	−0.687 to 0.203
>100%	0.4 ± 0.9	0.3 ± 0.6	0.081	−0.359 to 0.519
**Threshold**	**Competitive Games**			
50–60%	41.5 ± 29.5	44.7 ± 30.1	−0.129	−0.568 to 0.313
60–70%	35.6 ± 25.0	36.9 ± 22.0	−0.058	−0.496 to 0.381
70–80%	19.5 ± 13.9	19.5 ± 11.9	0.002	−0.436 to 0.440
80–90%	6.8 ± 4.6	6.7 ± 4.5	0.017	−0.422 to 0.455
90–100%	1.2 ± 0.7	1.3 ± 0.7	−0.167	−0.607 to 0.276
	**D20**			
	**C**	**NC**	**d**	**95% CI**
**Threshold**	**Training Sessions**			
50–60%	1.2 ± 1.3	5.4 ± 4.1	−1.086	−1.634 to −0.521
60–70%	0.8 ± 1.1	3.4 ± 3.2	−0.887	−1.398 to −0.358
70–80%	0.6 ± 0.9	1.5 ± 1.4	−0.522	−0.984 to −0.048
80–90%	0.4 ± 0.9	1.1 ± 1.3	−0.547	−1.011 to −0.069
90–100%	0.3 ± 1.0	0.3 ± 0.4	0.032	−0.406 to 0.470
>100%	1.7 ± 6.6	0.6 ± 1.2	0.158	−0.285 to 0.597
**Threshold**	**Competitive Games**			
50–60%	3.8 ± 2.6	4.9 ± 3.7	−0.258	−0.701 to 0.191
60–70%	2.4 ± 1.7	2.8 ± 1.7	−0.291	−0.735 to 0.160
70–80%	1.3 ± 1.5	1.7 ± 1.3	−0.265	−0.708 to 0.184
80–90%	0.7 ± 0.6	0.9 ± 0.9	−0.241	−0.683 to 0.207
90–100%	0.4 ± 0.5	0.6 ± 0.6	−0.288	−0.732 to 0.163

Values are mean ± SD. Effect sizes (d) and 95% CI: confidence intervals are shown. TD: total distance; D20: distance > 20 km·h^−1^; C: congested period; NC: non-congested period.

## Data Availability

Data is available upon request to the corresponding author.

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
