# Peer review of "Intensity vs. Volume in Professional Soccer: Comparing Congested and Non-Congested Periods in Competitive and Training Contexts Using Worst-Case Scenarios"

_sports, 2025, doi:10.3390/sports13030070_

Round 1

Reviewer 1 Report

Comments and Suggestions for Authors

Below are the major and minor weaknesses of the manuscript, categorized for clarity. Line numbers are provided for specific references.

Major Weaknesses

  1. Lines 10-14: The abstract fails to clearly justify why this study is necessary. While it mentions optimizing training load, it does not explain the gaps in prior research.
  2. Lines 88-90: Data were collected from only one professional club. This limits the generalizability of findings.
  3. Lines 93-98: How data were processed and validated is unclear.
    1. Were any outliers removed?
    2. Were any data quality checks performed?
  4. Line 133-136: The GPS system is described, but measurement error or inter-device variability is not mentioned.
  5. Lines 165-169: The manuscript uses repeated measures ANOVA but does not mention post-hoc corrections for multiple comparisons (e.g., Bonferroni or Holm correction). This raises concerns about Type I errors.
  6. Suggested Fix: Clearly state whether a correction method was applied, and if not, justify why.
  7. Line 171-173: The reported effect sizes are small or trivial in most cases, suggesting the findings might lack practical significance despite statistical significance.
  8. Line 229-231: The authors hypothesize that players with reduced playing time would accumulate higher WCS thresholds in training. However, the results do not support this, yet the discussion does not sufficiently explore why this hypothesis failed.
  9. Line 244-248: The discussion assumes that high-intensity training should mirror match demands but does not consider adaptation principles.

Minor Weaknesses

  1. Line 40-42: The manuscript references "maximal external load" but does not explicitly define how it differs from internal load.
  2. Line 115-120: The manuscript mentions tactical periodization but does not sufficiently explain how this model was structured in training.
  3. Line 252-254: Sentences are overly complex and contain unnecessary jargon (e.g., "microcycle periodization in elite football").
  4. Line 343-345: The reference formatting is inconsistent (some DOIs are missing, incorrect journal formatting).

Author Response

We would like to express our sincere gratitude for your thorough and constructive feedback on our manuscript. Your insightful comments have greatly contributed to improving the clarity and quality of our work. We have carefully considered all your suggestions and have made several revisions to address the points raised. We believe these modifications enhance the manuscript and provide a more robust presentation of our findings. Thank you once again for your valuable input.

Major Weaknesses

  1. Lines 10-14: The abstract fails to clearly justify why this study is necessary. While it mentions optimizing training load, it does not explain the gaps in prior research.

Thank you for your valuable feedback. We acknowledge that the initial abstract did not explicitly highlight the research gap. To address this, we have added a justification emphasizing the lack of studies examining the distribution of time spent within worst-case scenario (WCS) thresholds across different playing time groups and competitive periods. This addition reinforces the necessity of our study in providing insights for training periodization in professional soccer.

Line 10 : “Understanding the balance between intensity and volume during training and competition is crucial for optimizing players’ performance and recovery in professional soccer. While worst-case scenarios (WCS) are commonly used to assess peak match demands, little is known about how the time spent within WCS thresholds varies across congested and non-congested periods, especially when considering differences in playing time. This study examined the time spent at different percentages of WCS during congested and non-congested periods for lower and higher playing time players, throughout training sessions and matches.”

  1. Lines 88-90: Data were collected from only one professional club. This limits the generalizability of findings.

We acknowledge that data were collected from a single professional club, which may limit the generalizability of our findings. However, our study provides valuable insights into training load distribution within a real-world elite environment. Similar methodologies can be applied across different teams to validate our findings. Furthermore, in the limitation section at the end of the discussion we have already added the following sentence: “First, the findings of this study are based on data from a single professional team, which may limit their generalizability (line 353).”

  1. Lines 93-98: How data were processed and validated is unclear.
    1. Were any outliers removed?
    2. Were any data quality checks performed?

Thank you for your comment. We acknowledge the importance of detailing the data processing and validation steps. To ensure data quality, we applied several preprocessing steps: (1) raw data were inspected for inconsistencies or missing values, (2) extreme outliers, defined as values exceeding three standard deviations from the mean, were removed unless a contextual explanation was available, and (3) only sessions with complete tracking data were included in the analysis. We have now clarified these procedures in the manuscript.

Line 101 : “Data were obtained from the club throughout the 2024-2025 season as players were routinely monitored during training and matches. Therefore, ethics approval was waived. Nevertheless, to guarantee team and player confidentiality, all data were anonymized before analysis. To ensure data quality, all datasets were screened for missing values or inconsistencies. Extreme outliers (>3 standard deviations from the mean) were removed unless a contextual justification was identified. Only complete datasets from fully tracked sessions were included in the analysis. The research was conducted in accordance with the Declaration of Helsinki.”

  1. Line 133-136: The GPS system is described, but measurement error or inter-device variability is not mentioned.

Thank you for your insightful comment. We acknowledge the importance of reporting measurement error and inter-device variability. The Catapult Vector S7 10-Hz GPS system has been previously validated for reliability and accuracy in team sports contexts. To address your concern, we have now included this information in the manuscript.

Line 150 : “During all training and match sessions, players were equipped with a Catapult Vector S7 10-Hz GPS unit, integrated with a 100-Hz triaxial accelerometer microsensor (Vector S7, Catapult Innovations, Melbourne, Australia). This system has been validated for reliability in team sports [15]. The typical measurement error for GPS units in team sports is estimated to be approximately 1-2% for total distance and around 5% for high-speed running distance [16,17]. These values have been confirmed by a comparison between the Catapult Vector S7 system and an optical tracking system (Tracab 25-Hz), which revealed an excellent agreement for total distance (ICC = 0.974) and a good agreement for high-speed running distance (ICC = 0.766) [16]. To minimize inter-unit variability, each player used the same individual GPS device throughout the study, in line with best practice to ensure data consistency and reduce measurement errors [17].”

  1. Lines 165-169: The manuscript uses repeated measures ANOVA but does not mention post-hoc corrections for multiple comparisons (e.g., Bonferroni or Holm correction). This raises concerns about Type I errors.
  2. Suggested Fix: Clearly state whether a correction method was applied, and if not, justify why.

Thank you for your feedback. We confirm that Bonferroni-adjusted post-hoc tests were applied to control for multiple comparisons. However, we acknowledge that this was not explicitly stated in the statistical analysis section. To clarify, we have now revised the text to ensure transparency regarding our correction method.

Line 189 : "Prior to running the test, the assumption of sphericity was evaluated using Mauchly’s test. For the ANOVA, partial eta-squared (ηp²) was calculated as a measure of effect size. Effect sizes were interpreted as small (<0.06), moderate (0.06−0.15), or large (>0.15) [16]. Statistical significance was set at p<0.05.

Line 194 : In the event of significant main effects for period or playing time, or a significant interaction between these factors, Bonferroni-adjusted post-hoc tests were used to identify localized differences while controlling for multiple comparisons."

  1. Line 171-173: The reported effect sizes are small or trivial in most cases, suggesting the findings might lack practical significance despite statistical significance.

Thank you for your comment. We acknowledge that many of the reported effect sizes are small. However, in the context of elite soccer, even small differences in training load distribution can have meaningful implications for player adaptation, injury risk, and performance optimization. To address this, we have added a discussion point in the limitation section on the practical implications of our findings, emphasizing their relevance despite small effect sizes.

Line 366 : “Finally, although the effect sizes observed in our study were generally small, it is important to consider their practical implications within an elite soccer environment. Even minor differences in training load exposure can contribute to cumulative adaptations over a season, influence recovery strategies, and impact long-term player development.”

  1. Line 229-231: The authors hypothesize that players with reduced playing time would accumulate higher WCS thresholds in training. However, the results do not support this, yet the discussion does not sufficiently explore why this hypothesis failed.

  1. Line 244-248: The discussion assumes that high-intensity training should mirror match demands but does not consider adaptation principles.

We would like to thank the reviewer for their valuable feedback and insightful comments. We have carefully considered the points raised and made the necessary revisions to the discussion. Regarding the hypothesis about reduced playing time players accumulating higher WCS thresholds in training: We have expanded the discussion to provide a more thorough exploration of why this hypothesis was not supported by our results. Regarding the assumption that high-intensity training should mirror match demands without considering adaptation principles: We have revised the discussion to better account for adaptation principles, emphasizing that training should not only mirror match demands but also provide sufficient overload to promote progressive adaptation.

Line 374 : “While we hypothesized that players with reduced playing time would spend more time within various thresholds of the competitive WCS during training sessions, our results did not support this hypothesis, as no interaction between playing time and period was observed. This unexpected result suggests that the current training structure may not sufficiently differentiate workload based on playing time, highlighting the need for further investigation into alternative programming strategies.”

Line 305 : “This suggests that, despite their intended purpose, these sessions may not fully expose players to the highest intensities of match play. A possible explanation is that even when external loads appear to be matched, internal load responses—such as neuromuscular fatigue or metabolic strain—could vary, affecting overall adaptation.”

Line 325 : “Moreover, our findings suggest that simply mirroring match demands in training may not be sufficient for optimal adaptation. The principle of progressive overload must also be considered, ensuring that training loads progressively challenge players beyond match intensities over time to promote continued adaptations.”

Line 341 : “Future research should investigate whether modifying the intensity and duration of these drills could better align training stimuli with adaptation principles, particularly for players needing additional exposure to high-intensity efforts.”

Line 362 : “Integrating measures of neuromuscular fatigue and recovery status in future studies could help clarify whether current training designs adequately stimulate adaptation, particularly for players with reduced playing time.”

Line 369 : “Understanding how different training stimuli interact over time, rather than focusing solely on immediate match-like intensities, could offer a more holistic perspective on performance optimization.”

Minor Weaknesses

  1. Line 40-42: The manuscript references "maximal external load" but does not explicitly define how it differs from internal load.

Thank you for your comment. We have clarified the distinction between internal and external load in the revised manuscript. We agree that it is important to define these terms explicitly, as they are key concepts in understanding the demands of training and competition. We hope this addition provides a clearer explanation for the readers.

Line 43 : “External load refers to the physical demands placed on the body, such as total distance covered, speed, acceleration, or deceleration during training or competition, whereas internal load represents the physiological responses to those demands, typically measured through heart rate, rate of perceived exertion (RPE), or other physiological markers [1].”

  1. Line 115-120: The manuscript mentions tactical periodization but does not sufficiently explain how this model was structured in training.

Thank you for your valuable feedback. We have now provided a more detailed explanation of how the training model was structured according to tactical periodization principles. We hope this clarification helps to better contextualize the different phases of the training week in relation to match demands and objectives.

Line 125 : “The structure of this model follows the principles of tactical periodization, where each session is designed to target specific aspects of player performance. The day following the match (MD+1), the recovery session facilitates physical recovery to prepare players for the next sessions. On MD-4, neuromuscular development is emphasized through small-sided games, focusing on physical conditioning. On MD-3, the aerobic development session uses large-sided games to mimic match intensity. The speed-focused session on MD-2 addresses explosive power and acceleration through specific speed work. Finally, the tapering session on MD-1 aims to optimize reactivity and match-specific readiness through tactical exercises and small-sided games.”

  1. Line 252-254: Sentences are overly complex and contain unnecessary jargon (e.g., "microcycle periodization in elite football").

Thank you for your comment. We have simplified this sentence to make it clearer and remove any unnecessary jargon, ensuring it is more accessible while still conveying the intended meaning.

Line 301 : “Large-sided games, typically scheduled on MD-3, closely replicate match conditions and play a crucial role in high-intensity training during the week.”

  1. Line 343-345: The reference formatting is inconsistent (some DOIs are missing, incorrect journal formatting).

Thank you for pointing this out. We apologize for the inconsistencies in the reference formatting. We carefully reviewed all references and ensured that the correct journal formatting was applied, including the inclusion of any missing DOIs. The references have been revised for consistency throughout the manuscript. Unfortunaltely, the following reference does not have a DOI:” Buchheit, M.; Douchet, T.; Settembre, M.; Mchugh, D.; Hader, K.; Verheijen, R. The 11 Evidence-Informed and Inferred Principles of Microcycle Periodization in Elite Football. Sport performance and science reports 2024.”

Reviewer 2 Report

Comments and Suggestions for Authors

Thank you for the interesting manuscript. I have following questions/suggestions.

  1. When mentioning congested and non-congested in the abstract, it is unclear that they are about 'match' congestion. It is later described in the manuscript and it has to be clarified in the abstract.
  2. For the first sentence in the Introduction (line 34), is there a reference that can be added? What are considered primary objectives of training?
  3. In line 72-75, authors mention the results from 'unpublished data'. Will the data be published somewhere else? Can a reference be added or data be published/opened?
  4. According to Fig 1, all the data were collected in the beginning of the season. Can the results from this period be representative? Can the results change in mid-season or at the end of the season? I suggest authors discuss more on this aspect when describing Fig 1.
  5. Can authors disclose or discuss calibration data? Based on my experience, player tracking data and speed data (especially when extracted from tracking data) can be noisy and can be heavily affected by calibration methods. How was the calibrated data validated? Was it compared to other independently obtained data (e.g. comparing player speed from calibrated data to other kinds of measured speed data)?
  6. When interpreting effect sizes in line 172-173, is it arbitrary or can authors add references?
  7. In general, more descriptive captions for would be helpful. Currently captions are sometimes too general and do not help much with interpreting the data.
  8. When investigating the effect of 'Period x Playing Time', how exactly are 'period' and 'playing time' data combined? How can it be confirmed that it is the best way to combine 'period' and 'playing time' data?
  9. I could not find the data mentioned in line 209-211 in any table or figure. Where can those data be found? Also, when discussing the numbers from the data in the manuscript, I suggest adding specific references to figures or tables in parenthesis such as "(Table 1)".
  10. More descriptions on the numbers presented in tables and figures are needed. For example, in table 2, units for time is not mentioned and it is also unclear if they are 'total time' or 'time per session'. 
  11. There are many incidents of abbreviations being used before being explicitly defined. I suggest authors to double check that all abbreviations are explicitly defined before first use.
  12. It is not clear how the authors' conclusion can be interpreted as 'volume vs intensity'. I suggest making this connection clearer in the manuscript.

Author Response

We would like to express our sincere gratitude for your thorough and constructive feedback on our manuscript. Your insightful comments have greatly contributed to improving the clarity and quality of our work. We have carefully considered all your suggestions and have made several revisions to address the points raised. We believe these modifications enhance the manuscript and provide a more robust presentation of our findings. Thank you once again for your valuable input.

  1. When mentioning congested and non-congested in the abstract, it is unclear that they are about 'match' congestion. It is later described in the manuscript and it has to be clarified in the abstract.

We thank the reviewer for its comment.

Accordingly, we clarified: Line 17 : “Data were collected from a professional soccer team across a congested and non-congested match period”

  1. For the first sentence in the Introduction (line 34), is there a reference that can be added? What are considered primary objectives of training?

Thank you for your comment. In the first sentence of the Introduction, which discusses the primary objectives of training, we acknowledge that the main objectives often include improving physical performance, enhancing technical skills, and optimizing tactical awareness. We have now added relevant references to support this statement, particularly focusing on research that highlights these key objectives in soccer training.

Line 37 : “One of the primary objectives of training in sports is to replicate or even exceed competitive demands (physical, technical, tactical) to enhance athletes' performance [1–3]”

  1. In line 72-75, authors mention the results from 'unpublished data'. Will the data be published somewhere else? Can a reference be added or data be published/opened?

Thank you for your valuable feedback. We acknowledge that the publishing data would be beneficial. We therefore have added a supplementary table.

Line 79 : “Unpublished data from the authors indicate that the ten most frequently used players typically account for approximately 70% of the total playing time, while 16 players collectively share around 95% of it (Table S1)”

  1. According to Fig 1, all the data were collected in the beginning of the season. Can the results from this period be representative? Can the results change in mid-season or at the end of the season? I suggest authors discuss more on this aspect when describing Fig 1.

Thank you for your valuable comment. You are correct that all data were collected at the beginning of the season, and we acknowledge that this may influence the representativeness of our findings. We have now expanded our limitation section in the manuscript to address this limitation. Specifically, we have added this section:

Line 351 : “First, the findings of this study are based on data from a single professional team at a specific time of the year, which may limit their generalizability. Future research should aim to replicate these results in different contexts, such as with youth players focused on development rather than competitive outcomes, or in different country with varying methodologies. Furthermore, the study should be replicated during mid-season or at the end of the season.”

  1. Can authors disclose or discuss calibration data? Based on my experience, player tracking data and speed data (especially when extracted from tracking data) can be noisy and can be heavily affected by calibration methods. How was the calibrated data validated? Was it compared to other independently obtained data (e.g. comparing player speed from calibrated data to other kinds of measured speed data)?

Thank you for raising this important point. We acknowledge that player tracking and speed data can be influenced by calibration methods, potentially affecting data reliability. The Catapult Vector S7 10-Hz GPS system has been previously validated for reliability and accuracy in team sports contexts. To address your concern, we have now included this information in the manuscript.

Line 150 : “During all training and match sessions, players were equipped with a Catapult Vector S7 10-Hz GPS unit, integrated with a 100-Hz triaxial accelerometer microsensor (Vector S7, Catapult Innovations, Melbourne, Australia). This system has been validated for reliability in team sports [15]. The typical measurement error for GPS units in team sports is estimated to be approximately 1-2% for total distance and around 5% for high-speed running distance [16,17]. These values have been confirmed by a comparison between the Catapult Vector S7 system and an optical tracking system (Tracab 25-Hz), which revealed an excellent agreement for total distance (ICC = 0.974) and a good agreement for high-speed running distance (ICC = 0.766) [16]. To minimize inter-unit variability, each player used the same individual GPS device throughout the study, in line with best practice to ensure data consistency and reduce measurement errors [17].”

  1. When interpreting effect sizes in line 172-173, is it arbitrary or can authors add references?

Thank you for your comment. The interpretation of effect sizes was based on established guidelines in the sports science literature. We have now included appropriate references to support the thresholds used in our analysis. Specifically, we refer to Cohen (1988) for general effect size interpretation.

Line 197 : “Effect sizes were interpreted as trivial (<0.2), small (0.2−0.5), moderate (0.5−0.8), or large (>0.8) [16].”

  1. In general, more descriptive captions for would be helpful. Currently captions are sometimes too general and do not help much with interpreting the data.

Thank you for your feedback. We have revised the figure captions to provide more detailed descriptions that better guide the reader in interpreting the data. 

Line 142 : Figure 1. Study flowchart illustrating the characteristics of congested and non-congested periods. The number of training sessions and games are shown. The total number of players, the total number of different starters, and the number of changes in the starting eleven throughout all the competitive games of each period are shown.

Line 260 : Figure 2. Comparison of the total time spent in all worst-case scenario (WCS) for total distance (TD) thresholds during training sessions between two groups based on playing time: PT 1-10 (players with the ten highest playing time) and PT 11-20 (players with the ten lowest playing time). Significant differences (p < 0.05) are indicated by *.

Line 267 : Figure 3. Comparison of the total time spent in all worst-case scenario (WCS) for distance covered >20km/h (D20) thresholds during training sessions between the two playing time groups: PT 1-10 (players with the ten highest playing time) and PT 11-20 (players with the ten lowest playing time). Significant differences (p < 0.05) are indicated by *.

  1. When investigating the effect of 'Period x Playing Time', how exactly are 'period' and 'playing time' data combined? How can it be confirmed that it is the best way to combine 'period' and 'playing time' data?

Thank you for your comment. In our analysis, we investigated the interaction between period (congested vs. non-congested) and playing time (PT 1-10 vs. PT 11-20) using a two-way repeated measures ANOVA. This statistical approach allowed us to assess both the main effects of each factor (i.e., how playing time and period independently influence the dependent variables) and their interaction effect (i.e., whether the effect of playing time on WCS thresholds varies depending on the period).

The combination of these two factors followed a within-subjects design, as each player experienced both congested and non-congested periods over the course of the study. This method is commonly used in sports science research when analyzing repeated measures over time and ensures that inter-individual variability is accounted for.

To confirm that this was an appropriate approach, we ensured that:

  • Data were structured appropriately to reflect repeated observations per player across different periods.
  • The assumption of sphericity was tested, and corrections were applied if necessary.
  • Alternative methods (e.g., mixed models) were considered, but a two-way repeated measures ANOVA was preferred due to its ability to clearly test interaction effects while maintaining statistical power with our sample size.

  1. I could not find the data mentioned in line 209-211 in any table or figure. Where can those data be found? Also, when discussing the numbers from the data in the manuscript, I suggest adding specific references to figures or tables in parenthesis such as "(Table 1)".

Thank you for your observation. You are correct that the data mentioned in lines 209-211 were not initially included in any figure or table. We have now added these data to the result section to ensure clarity and transparency.

Line 243 : Figures 2 and 3 show the results for WCS TD and WCS D20 for the playing time during training sessions. The playing time parameter (PT 1-10 vs. PT 11-20) did not show any significant difference for WCS TD (Table 1). However, for WCS D20, playing time differentiated the time spent between 80-90% (p=0.023, d=-0.471, 95% CI=-0.916 to -0.026), and 90-100% (p=0.042, d=-0.595, 95% CI=-1.204 to 0.015) during training sessions with greater values for PT 11-20 (Table 1). For WCS D20 during competitive games, greater values were obtained for PT 11-20 were highlighted between 50-60% (7.65 ± 3.07 vs 2.08 ± 11.36, p=0.002, d=1.513, 95% CI=0.462 to 2.564), 60-70% (3.56 ± 1.59 vs 2.13 ± 1.49, p=0.048, d=0.714, 95% CI=-0.033 to 1.462), and 70-80% ( 2.20 ± 1.35 vs 1.20 ± 1.19, p=0.029, d=0.688, 95% CI=0.031 to 1.345) while no difference was highlighted between 80-90% (1.05 ± 1.01 vs 0.78 ± 0.69, p=0.689, d=0.103, 95% CI=-0.117 to 0.345) and 90-100% (0.80 ± 0.68 vs 0.48 ± 0.35, p=0.173, d=0.183, 95% CI=-0.073 to 0.645).

Additionally, we have revised the manuscript to include explicit references to tables and figures when discussing numerical results. This should improve readability and help readers quickly locate the corresponding data.

We appreciate your feedback, which has helped enhance the clarity and organization of our manuscript.

  1. More descriptions on the numbers presented in tables and figures are needed. For example, in table 2, units for time is not mentioned and it is also unclear if they are 'total time' or 'time per session'. 

Thank you for your valuable feedback. We have carefully reviewed all tables and figures to ensure clarity and completeness.

We have added the term “total” to all figures and tables and used plural for training sessions and competitive games to ensure clarity.

  1. There are many incidents of abbreviations being used before being explicitly defined. I suggest authors to double check that all abbreviations are explicitly defined before first use.

Thank you for your careful review. We have thoroughly checked the manuscript to ensure that all abbreviations are explicitly defined before their first use. Any instances where abbreviations were used prematurely have been corrected. We appreciate your suggestion, as it improves the clarity and readability of the manuscript.

  1. It is not clear how the authors' conclusion can be interpreted as 'volume vs intensity'. I suggest making this connection clearer in the manuscript.

Thank you for your valuable feedback. We have clarified the connection between our findings and the "volume vs. intensity" interpretation within the discussion.

Round 2

Reviewer 1 Report

Comments and Suggestions for Authors

Thank you for implementing my comments